

# Combined transcriptome and proteome analysis of yak PASMCs under hypoxic and normoxic conditions

Lan Zhang[1], Yiyang Zhang[1,2,3], Juan Zhou[1], Yifan Yao[1,2,3], Rui Li[1,2,3], Manlin Zhou[1], Shuwu Chen[1,2,3], Zilin Qiao[1,2,3] and Kun Yang[1,2,3]

[1] Life Science and Engineering College, Northwest Minzu University, Lan, China
[2] Biomedical Research Center, Northwest Minzu University, Lan Zhou, China
[3] Gansu Tech Innovation Center of Animal Cell, Lan Zhou, China

Corresponding author
Kun Yang, 186152592@xbmu.edu.cn

## ABSTRACT

**Background**. Yaks are animals that have lived in plateau environments for generations. Yaks can adapt to the hypoxic plateau environment and also pass this adaptability on to the next generation. The lungs are the most important respiratory organs for mammals to adapt to their environment. Pulmonary artery smooth muscle cells play an important role in vascular remodeling under hypoxia, but the genetic mechanism underpinning the yak's ability to adapt to challenging plateau conditions is still unknown.

**Methods**. A tandem mass tag (TMT) proteomics study together with an RNA-seq transcriptome analysis were carried out on pulmonary artery smooth muscle cells (PASMCs) that had been grown for 72 hours in both normoxic (20% O2) and hypoxic (1% O2) environments. RNA and TP (total protein) were collected from the hypoxic and normoxic groups for RNA-seq transcriptome sequencing and TMT marker protein quantification, and RT-qPCR validation was performed.

**Results**. A total of 17,711 genes and 6,859 proteins were identified. Further, 5,969 differentially expressed genes (DEGs) and 531 differentially expressed proteins (DEPs) were identified in the comparison group, including 2,924 and 186 upregulated genes and proteins and 3,045 and 345 down-regulated genes and proteins, respectively. The transcriptomic and proteomic analyses revealed that 109 DEGs and DEPs were highly positively correlated, with 77 genes showing the same expression trend. Nine overlapping genes were identified in the HIF-1 signaling pathway, glycolysis / gluconeogenesis, central carbon metabolism in cancer, PPAR signaling pathway, AMPK signaling pathway, and cholesterol metabolism (PGAM1, PGK1, TPI1, HMOX1, IGF1R, OLR1, SCD, FABP4 and LDLR), suggesting that these differentially expressed genes and protein functional classifications are related to the hypoxia-adaptive pathways. Overall, our study offers abundant data for further analysis of the molecular mechanisms in yak PASMCs and their adaptability to different oxygen concentrations.

# INTRODUCTION

Low oxygen content is one of the main characteristics of a plateau environment. Some studies have shown that plains animals that rush into the plateau can develop heart

disease and "animal chest disease." Yaks living at high altitudes for generations can adapt to the hypoxic environment that exists at high altitudes; these animals do not develop pulmonary hypertension and can hereditarily pass their adaptability on to the next generation. The lungs are the most important organs in the respiratory system, and can adapt to the external environment through a series of physiological changes. Studies have shown that the sensitivity of animal lung tissue to hypoxia is closely related to the content of smooth muscle on the pulmonary vascular wall; the higher the smooth muscle content, the higher the sensitivity to hypoxia (*Tucker, McMurtry et al., 1975*). Vascular smooth muscle cells (VSMCs) maintain high plasticity in the pulmonary blood vessel system. Under normal physiological conditions, these cells maintain low proliferation, low migration, strong contractile force, and express a set of specific cytoskeletal and contractile proteins. Under different environmental conditions, muscle cells re-enter the cell cycle and transform from the contractile phenotype in the differentiated state to the secretory phenotype in the dedifferentiated state. In the dedifferentiated state, PASMCs have low contractile ability and a strong ability to proliferate, migrate and secrete extracellular matrix (ECM). Therefore, further research on the molecular mechanisms of PASMC dedifferentiation is important for understanding and preventing cardiovascular diseases. PASMCs are involved in apoptosis, antioxidant activity, vascular composition, and other processes (*Huetsch, Suresh & Shimoda, 2019*). The interaction between VSMCs and other cells causes a continuous increase of vasoconstrictive forces and abnormal vascular proliferation. PASMCs involved in other diseases play a crucial part in preserving the balance of pulmonary circulation (*Tuder et al., 2013*; *Gao, Chen & Raj, 2016*). However, the current data to justify the adaptive growth of plateau animals are insufficient.

In this study, RNA-seq and TMT technology were used for transcriptome sequencing and proteomic analysis of PASMCs under hypoxic and normoxic conditions. A better understanding of the differentially expressed genes and regulatory pathways of PASMCs will provide a foundation for further research into the molecular circadian mechanism of PASMCs growth and a comprehensive understanding of the specificity of PASMC in yaks.

## MATERIAL AND METHODS

### Animal ethics
Clinically healthy animals were identified through a physical evaluation and biochemical serum analysis prior to slaughter. The People's Republic of China's Animal Ethical Regulations and Guidelines were followed in all experimental procedures. The Animal Ethics Committee of the Northwest University for Nationalities also approved this study.

### Culturing yak PASMCs
Samples of yak pulmonary arteries were collected from a slaughterhouse in Xining, Qinghai Province. Part of these arteries were then kept in liquid nitrogen, and the other part were stored in normal saline at about 25 °C. Data were collected according to the PASMC culture method of *Zhang et al. (2021)*. The pulmonary arteries were washed three times with saline and PBS prior to culturing. A scalpel was then used to gently scrape away the endothelial cells, cutting the tissue into three 1 mm pieces. These tissue samples were then

placed in 1 mg/mL DMEM/F12 medium. The cells were centrifuged at 1,000 RPM for 5 min, harvested, and then resuspended in whole DMEM/F12 medium containing 20% fetal bovine serum (FBS). When the principal cells grew to 90% confluence, they were neutralized with 0.25% trypsin + 0.01% EDTA and DMEM/F12 containing 15% FBS. Cells were then divided into hypoxic and normoxic groups and cultured using the adherent subculture approach with three replicates in each group.

## RNA extraction, and library construction and sequencing

The cells (three replicates per group) were gathered and cultured every 72 h using TRIzol reagent (six-well plate with 80% cell density). The supernatant was discarded after 5 min of centrifugation at 2,000 RPM. We mixed one mL TRIzol into the cell sediment, let it sit for 5 min, and then transferred it to a new 1.5 mL centrifuge tube. Agarose gel electrophoresis was performed to determine RNA integrity and the presence of DNA contamination. High-throughput sequencing was used for library development and verification. Sequencers were used to capture fluorescence indicators and a laptop software program was used to convert the optical alerts into sequencing peaks to obtain the sequence statistics of the examined fragment. Data were collected according to the sequencing method of *Pertea et al. (2015)*. In this step, clean data (clean reads) were obtained for high quality analysis. The reference genome and gene mannequin annotation documents were downloaded from the NCBI genome website. StringTie (1.3.3B) was used for new gene prediction. The FPKM of each gene was then calculated based on length and read counts were determined. A differential expression analysis between the two comparison combinations (two biological replicates per group) was performed using the DESeq2 software (1.16.1). The resulting *P*-value was adjusted to control for false discovery using the Benjamini Hochberg's procedure. For differentially expressed genes, *P* values <0.05 with a threshold of | log2foldchange | were considered significantly differentially expressed. The statistical power of this experimental design, calculated in RNASeqPower (http://www.bioconductor.org/packages/release/bioc/html/RNASeqPower.html) as "rnapower (821.7883585, $n = 3.3$, cv $= 0.039387208$, effect $=2$, alpha $=0.05$)" is 1 (*Therneau, Hart & Kocher, 2022*).

## Expression profiles determined by RT-qPCR

We chose five DEGs from the differentially expressed genes in key regulatory pathways for RT-qPCR verification (Table 1). To obtain a supernatant, cells from each group were gathered and centrifuged at 2000 RPM for 5 min. TRIzol (one mL) was delivered to the cell precipitate, which was then combined and let stand for 5 min. The cells were then used for whole RNA extraction. The cDNA was obtained using a reverse transcription kit. A total of 2 μL of reverse transcription primer (0.5 μg/μL) and 2μg total RNA were added to the PCR tubes, with Rnase-free $H_2O$ supplemented to 11 μL. After mixing, the primer and template were centrifuged, incubated at 73.5 °C for 7 min, and then immediately placed in an ice bath and the reverse transcription primer and template were annealed. The above system was reacted in a water bath at 43.5 °C for 1 h, followed by a water bath at 73.5 °C for 3 min to inactivate the RT enzyme. The resulting reverse transcription product cDNA

**Table 1  Primer information for RT- qPCR.**

| Gene | Primer sequence | | Product size (bp) |
|------|-----------------|---|-------------------|
| OSMR | F:ACTGCCTTCCTACACTTTAT | R:GTGAGTCTTGAGTTACTTGC | 95 |
| EREG | F:ACCTGGTAGACATGAGTGAA | R:CAGCAACTATGACAAGGAAC | 156 |
| CSF1 | F:AGGAGGTGTCGGAGAACTGT | R:GTCTTTGAAGCGCATGGTAT | 206 |
| SPP1 | F:CGATGATGATAACAGCCAGGAC | R:CGTAGGGATAAATGGAGTGAAA | 166 |
| PTGS2 | F:GTTTTCTGCTGAAGCCCTAT | R:AAAACCTACTTCTCCACCGA | 263 |

**Table 2  RT-qPCR system.**

| Reagent | Quantity added per tube |
|---------|-------------------------|
| 5x RT buffer | 5 µl |
| 10 mM dNTPs | 2 µl |
| Rnasin (40 U/µL) | 0.5 µl |
| M-MLV-RTase (200 U/µl) | 0.5 µl |
| RNase-Free H2O | 6 µl |

**Table 3  RT-qPCR reaction system.**

| Reagent | Quantity added per tube |
|---------|-------------------------|
| SYBR premix ex taq | 6.0 µL |
| Forward primer (5 µM) | 0.5 µL |
| Downstream primers (5 µM) | 0.5 µL |
| Template (reverse transcription product) | 1.0 µL |
| RNase-Free H2O | 4.0 µL |

was stored at −20 °C for later use (Table 2). As show in Table 3, a two-step proportional response machine (12 µL) was used for RT-qPCR detection with the cycles set as follows: 95 °C for 1 min, 95 °C for 10 s, 60 °C for 40 s, for forty total cycles. After the reaction, the solubility curves were analysed. ACTB was used as an interior reference gene and the relative expression ranges of every gene were calculated using of the $2^{-\Delta\Delta Ct}$ method.

## Total protein extraction and TMT quantitative proteomic analysis

Proteins (six-well plate with 80% cell density) were extracted with SDT lysis. After extraction, one mL of the SDT lysate (P0013G; Beyotime, Haimen, China) was added to the cell samples. Ultrasound was then used on the cells and then the cells were boiled in a water bath for 15 min and then centrifuged for 15 min at 14,000 RPM. The BCA protein Assay Kit was used to measure the supernatant (P0012; Beyotime, Haimen, China). A total of 25 µg of protein from each group was added to 6X loading buffer and then boiled for 5 min before SDS-PAGE gel separation. Protein bands were visualized with the aid of Coomassie Blue R-250 staining. FASP enzymolysis was performed on all samples. A total of 100 µg of peptide was then collected from each sample and labelled according to the TMT labelling kit (90064CH; Thermo Fisher Scientific, Waltham, MA, USA) and the labelled peptides were mixed and graded. The column was balanced with a 100% A

solution (10 mmol $L^{-1}$ HCOONH4, 5% ACN, pH = 10.0). During the elution process, the absorption value was 214 nm and approximately 40 eluted components were collected every 1 min. The samples were lyophilised and re-dissolved in 0.1% FA and divided into 10 parts. Each pattern was separated using an EASY nLC machine with a 1 nanolitre per minute flow rate. The original data were obtained and the blank values were removed and the trusted proteins were screened according to the ScoreSequestHT>0 and Uniquepeptide ≥1 conditions. Based on the screened trusted proteins, DEP screening was carried out with FC ≥1.3 or ≤0.78 and $P < 0.05$. The Blast2GO software program was then used to perform the GO enrichment analysis of differentially expressed proteins, identifying the locations of the DEPs from three aspects: organic processes, molecular features, and cell components. The enriched DEPs were then aligned with the Kyoto Encyclopedia of Genes and Genomes (KEGG) pathways and the functional area annotation evaluation of the DEPs was performed using the Interpro database.

### Combined transcriptome and proteomics analysis

Differentially expressed proteins and mRNA were extracted from the combined proteome and transcriptome analysis, and the results were subjected to a correlation clustering analysis and GO/KEGG enrichment analysis.

## RESULTS

### Gene expression distribution

The sequencing depth of this experiment was 6G with three replicates per group. Due to the effects of sequencing depth and gene length, the RNA-seq gene expression values were recorded using FPKM rather than examine counts. Box plots were used to show the distribution of gene expression in different samples (Fig. 1). Based on this figure, the repeatability between parallel samples was relatively good.

### Intersample correlation

The closer the correlation coefficient of the gene expression of the samples is to 1, the greater the similarity of expression patterns between samples. We required $R^2$ to be greater than 0.8 between duplicate biological samples, or are interpretation of the sample or a re-run of the experiment was necessary. The Encode software program identified a Pearson correlation coefficient rectangle ($R^2$) greater than 0.92 under ideal sampling and experimental conditions. According to the FPKM values of all the genes in each sample, the correlation coefficients of the samples within and between groups were calculated, and a heat map was created to visually depict the sample similarities within groups and the sample differences between groups. The greater the correlation coefficient, the closer the expression sample (Fig. 2). PASMCs had a strong correlation between biological duplicates under hypoxic and normoxic conditions, indicating the reliability of the sampling and sequencing.

### Differential analysis of transcriptome sequencing gene expression

In this study, the two groups of samples were compared under hypoxic and normoxic conditions. Software combinations were compared to derive the differences in genetic

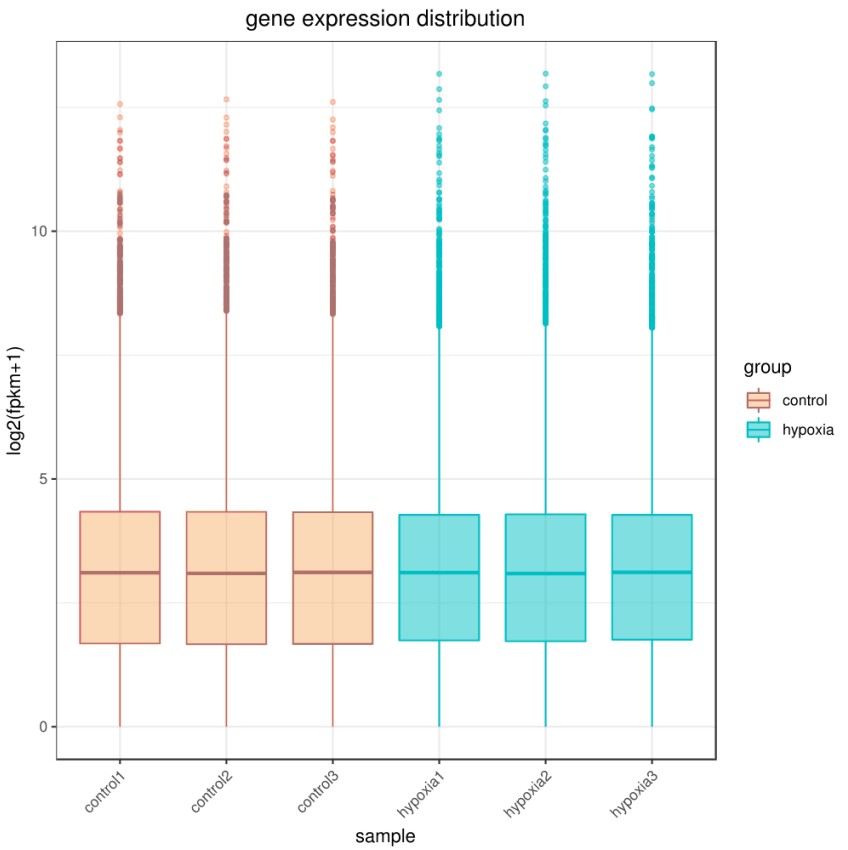

**Figure 1** **Box diagram of gene expression level distribution in samples.** In the figure, the abscissa is the sample name and the ordinate is log2 (FPKM+1).

screening and the threshold for DESeq2 (|log2(FoldChange)|>0 & padj <= 0.05). A total of 17,711 genes were expressed and 5,969 DEGs were identified in the comparison of the two sample, which 2,924 were up-regulated genes and 3,045 were down-regulated genes (Fig. 3A). A cluster diagram of the DEPs (Fig. 3B) revealed that the samples under the two conditions had good biological repeatability. The ClusterProfile software program was used for the GO enrichment evaluation of the differentially expressed gene sets. Generally, a Padj <0.05 indicated significant enrichment. In Fig. 3C, the top 30 terms were chosen to generate a bar chart and histogram of the organic processes, cell components, molecular functions, and down-regulation of differential genes. A KEGG evaluation of all the differentially expressed genes was carried out with a Padj <0.05 indicating significant enrichment, and the top 20 KEGG pathways were chosen to generate a bar chart (Fig. 3D). The DEGs were mainly enriched in the biosynthesis of amino acids, microRNAs in cancer, the phagosome, human papillomavirus infection, steroid biosynthesis, Alzheimer's disease, colorectal cancer, fatty acid metabolism, the MAPK signalling pathway, Epstein-Barr virus infection, apoptosis, the AGE-RAGE signalling pathway, ECM-receptor interaction, cellular senescence, the PI3K-Akt signalling pathway, pancreatic cancer, and the lysosome.
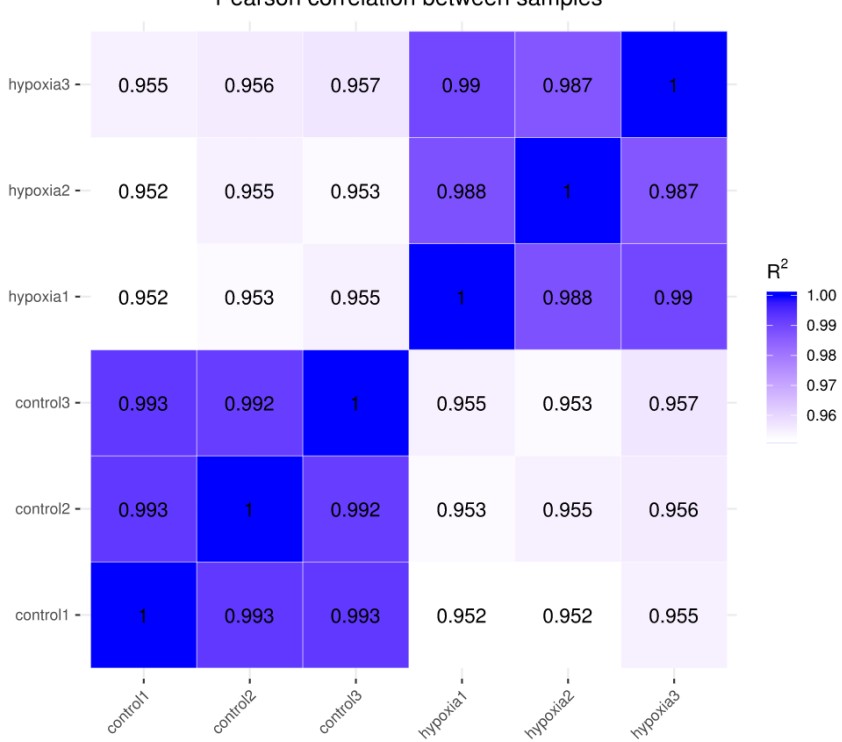

**Figure 2 Heat map of correlation between samples.** The closer the correlation coefficient is to 1, the higher the similarity of expression patterns between samples.

Research pathways identified from relevant literature or identified in the study were screened, followed by random screening for verification (Fig. 4A). A correlation analysis was used to determine dependability: the nearer $R^2$ is to 1, the more suitable the correlation and the more consistent and dependable the results between RT-qPCR and RNA-seq (Fig. 4B and Fig. 4C).

## Proteomic analysis

Based on the trusted proteins screened in the two groups, DEPs screening was performed according to FC Foldchange $\geq 1.3$ or $\leq 0.78$ and $P < 0.05$. A total of 6859 proteins were identified, including 531 DEPs (186 up-regulated and 345 down-regulated) in the comparison group (Fig. 5A). To further understand the biological significance of the DEPs identified under hypoxia and normoxia, DEP enrichment was analysed using GO functional annotation and KEGG pathway enrichment. The GO enrichment analysis revealed (Fig. 5B) that the differentially expressed proteins were enriched in several categories: extracellular space, integral component of plasma membrane, cell surface, serine-type endopeptidase inhibitor activity, heparin binding, calcium ion binding, negative regulation of endopeptidase activity, aging, and neutrophil degranulation. The KEGG analysis also revealed (Fig. 5C) significant DEP enrichment in the biosynthesis of secondary metabolites, the lysosome, complement and coagulation cascades, the HIF-1 signalling

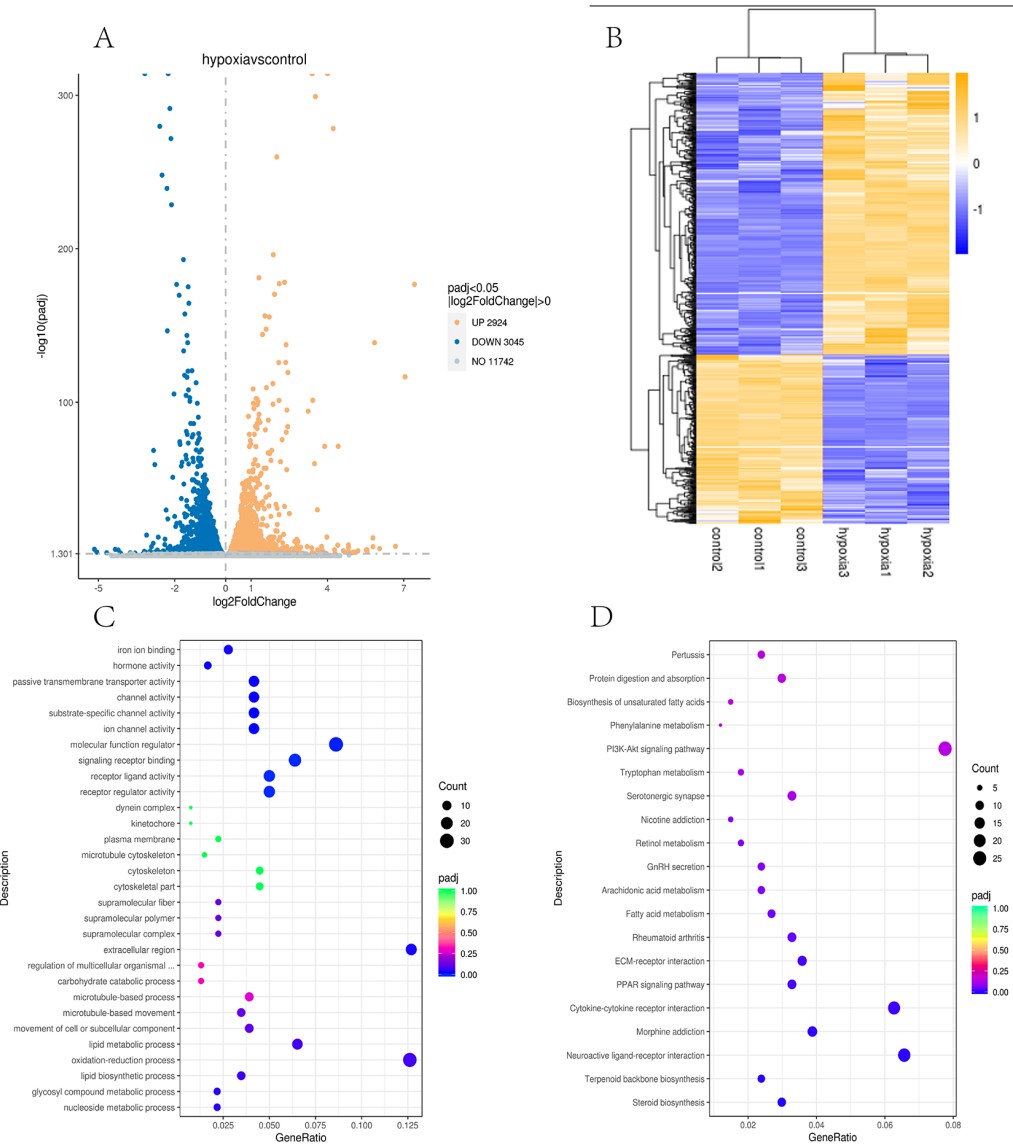

**Figure 3** **Transcriptomic analysis of PASMCs under hypoxic and normoxic conditions.** (A) Differential gene volcano plot; Up-regulated genes are indicated by yellow dots and down-regulated genes are indicated by blue dots. (B) Cluster heat map of differentially expressed genes; The abscissa is the sample name, and the ordinate is the normalized FPKM value of the differential gene. (C) The abscissa is the ratio of the number of differential genes annotated to the GO Term to the total number of differential genes, and the ordinate is GO Term. (D) The abscissa is the ratio of the number of differential genes annotated to the KEGG pathway to the total number of differential genes, and the ordinate is the KEGG pathway.

pathway, glycolysis/gluconeogenesis, cholesterol metabolism, central carbon metabolism in cancer, the PPAR signalling pathway, cytokine-cytokine receptor interaction and other glycan degradation.

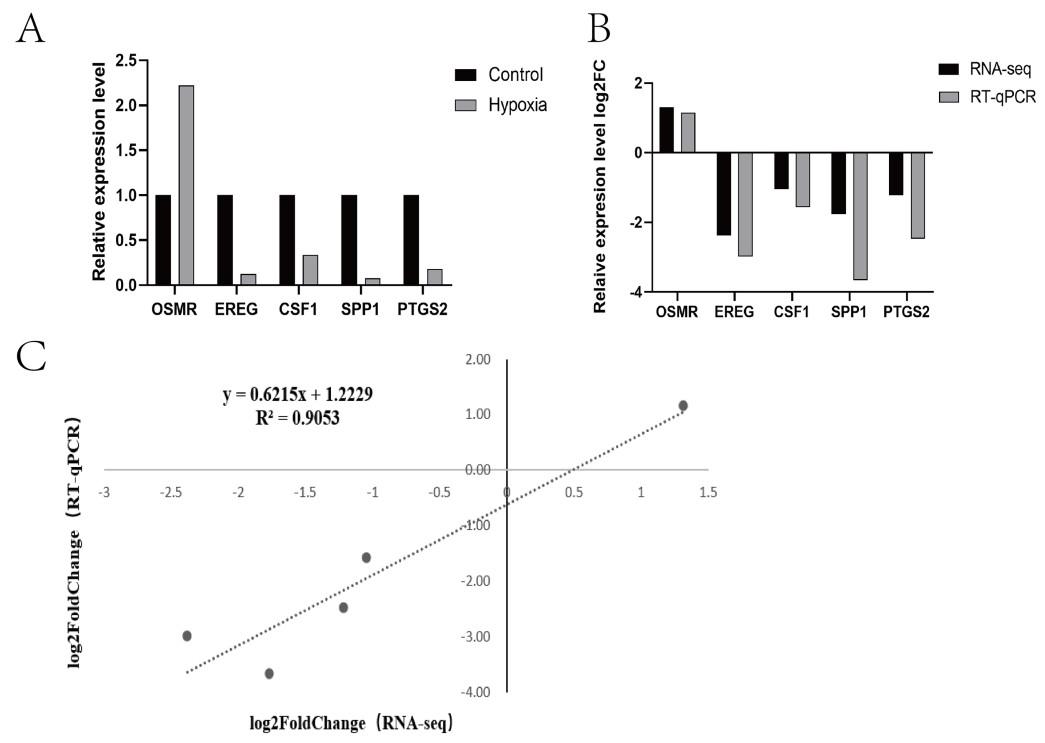

**Figure 4** **Expression patterns of five differentially expressed genes.** (A) Transcriptome sequencing was verified by RT-qPCR. (B) Expression patterns of five differentially expressed genes (RT-qPCR and RNA-seq) (C) Dot plot of correlation between RNA-Seq and RT-qPCR.

## Combined transcriptome and proteome analysis

Transcriptome and proteome data were analysed and a Venn diagram of the DEPs and genes in the hypoxia and normoxia groups was created (Fig. 6A). A total of 109 common differential genes were identified between PASMCs under hypoxic and normoxic conditions. Significantly different proteins and genes were correlated to highlight the significance level between the different omics. The absolute value of log2FC of the transcriptome was greater than 0, while that of the proteome log2FC was greater than 0.263, indicating an obvious trend (Fig. 6B). The horizontal and vertical coordinates in the figure represent the multiple differences (log2 value) in protein/gene expression levels in each comparison group. Yellow dots (opposite) indicate opposite trends in protein and transcriptome expression; blue dots (same) indicate the same trend in expression at the protein and transcriptome levels. The transcriptome had very little correlation with the proteome. Hierarchical clustering of differential proteins and associated transcriptome data was performed using a heat map (Fig. 6C). Each row in the figure represents a differentially intersected gene. As shown in the figure, the expression stages of proteins and related transcripts were inconsistent at each the quantifiable and enormous distinction levels, and sure variations have been observed in the expression diploma and expression trend. In addition to post-transcriptional regulation, the gene transcription and expression may have a spatial and temporal order. In the transcriptome and proteome association analysis results,

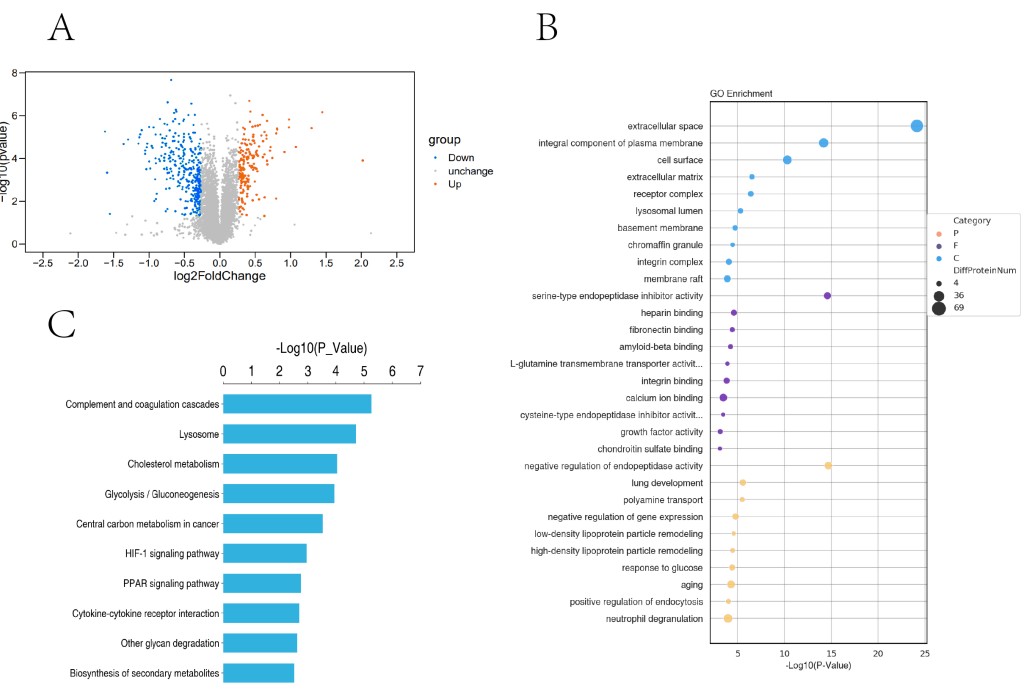

**Figure 5** **Proteomic analysis of PASMCs under hypoxic and normoxic conditions.** (A) Proteomic volcano plot. The abscissa is the multiple of difference (logarithmic transformation based on 2). The ordinate is the *p*-value (log change based on 10). (B) Statistical bubble chart of GO terms with significant enrichment. The ordinate represents the GO function name, and the abscissa represents the enrichment significance *p*-value; The color of the circle indicates the functional classification of GO (P: Biological Process, F: Molecular Function, C: Cellular Component), and the size of the circle indicates the number of differential proteins contained therein. (C) Bar chart of statistically significant enriched KEGG pathways. The ordinate represents the KEGG pathway, and the abscissa represents the *p*-value of enrichment significance.

enriched biological process (BP), cellular component (CC) and molecular function (MF) items were shown using a GO enrichment analysis (Figs. 7A, 7B and 7C), with the abscissa representing the significantly enriched functional items and the ordinate representing the significance level FDR (negative log value based on 10). In the figure, "same" represents the enrichment analysis results of genes with the same expression trends in the proteome and transcriptome while "opposite" represents the enrichment analysis results of genes with opposite proteome and transcriptome expression trends. "Protein only" indicates the enrichment analysis results of differentially expressed genes in the proteome with no change in the transcriptome. "Transcript_only" indicates that there was no change in the proteome and the enrichment analysis results of differentially expressed genes in the transcriptome. Genes with similar expression trends in the proteome and transcriptome were mainly enriched in the glycolytic process, cholesterol homeostasis, positive regulation of inflammatory response, negative regulation of gene expression, cell migration, positive regulation of gene expression, positive regulation of cell population profile, inflammatory response, sodium transport, and epithelial cell differentiation. Genes with the same expression trends were mainly concentrated in proteolysis, signal transduction, glucose metabolic processes, fatty acid metabolic processes, positive regulation of I-kappab

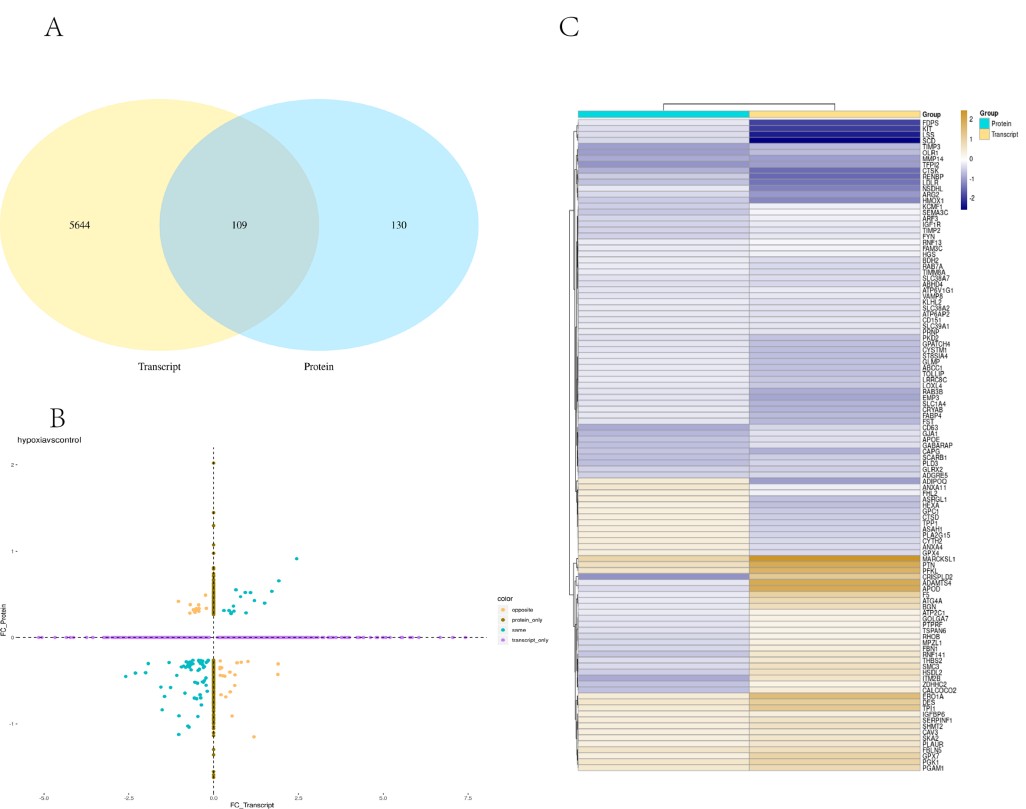

**Figure 6 Combined analysis of PASMCs under hypoxic and normoxic conditions.** (A) Statistics of correlation number Venn diagram (hypoxia *vs* control). (B) Correlation analysis between the ex pression levels of all significantly different proteins and their associated genes. In the figure, the horizontal and vertical coordinates are the transcriptome and protein expression levels (logarithmic transformation based on 2). (C) Results of cluster analysis (Hypoxia *vs* control).

kinase/NF-kappab signalling and cell adhesion. KEGG is the most commonly used authoritative database for pathway analysis. Pathway enrichment analysis results were similar to those of the GO enrichment analysis. Using the pathway as the unit and all known genes of all genes or species on the chip as the background, the significance level of the gene enrichment of each pathway was analysed and calculated using the Fisher's exact test. The metabolic and sign transduction pathways were determined to be considerably affected. Through KEGG pathway annotation and enrichment analysis, the significance levels of the top 10 pathways from the proteome and transcriptome analysis results were analysed (Fig. 7D). The abscissa represents the significantly enriched pathway, and the ordinate represents the enriched value stage FDR (negative log value based on 10). Amino acids, carbon metabolism, endocytosis, and glycolysis/gluconeogenesis were enriched by genes with the same expression trends in the proteome and transcriptome. In addition, proteoglycans in cancer, the phagosome, the HIF-1 signalling pathway, autophagy-animal, ovarian steroidogenesis, and other pathways were found to be enriched. Genes with opposite expression trends were mainly enriched in the lysosomal pathway.

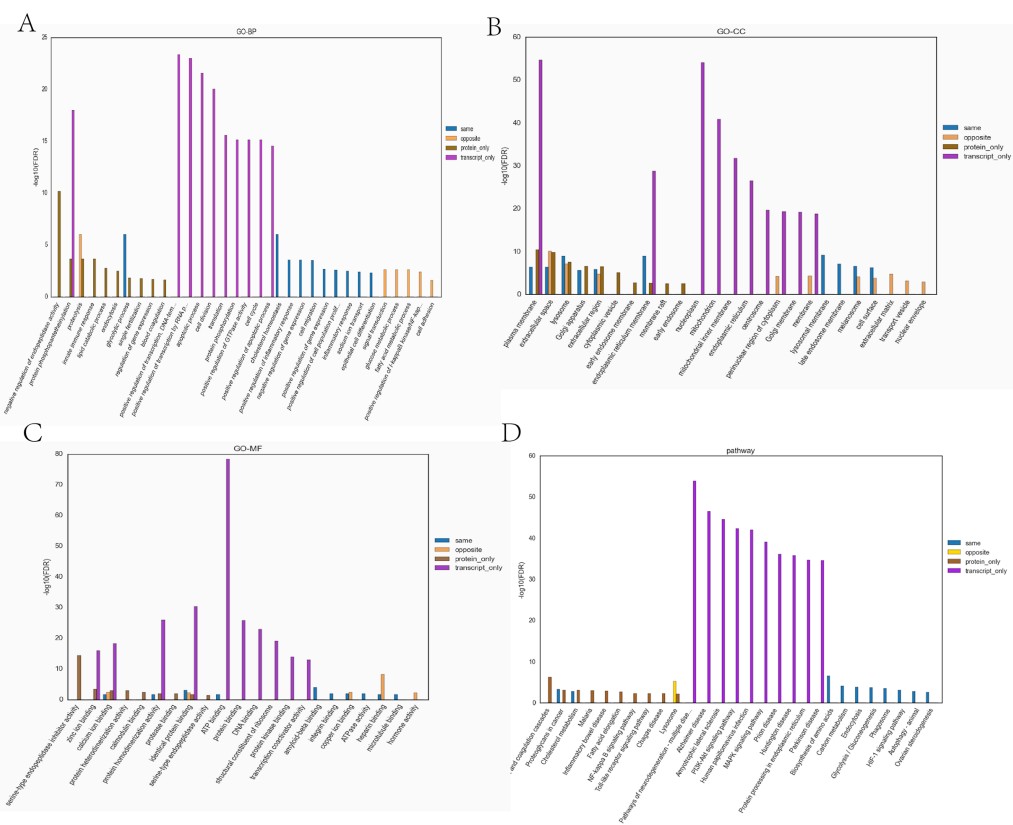

**Figure 7 The top ten significantly enriched GO terms and KEGG comparison chart (Top 10) (Hypoxia vs control).** (A) Comparative plots of enrichment in BP (Biological Process). (B) Comparative plots of enrichment in CC (Cellular Component). (C) Comparative plots of enrichment in MF:(Molecular Function). (D) Bar chart of statistically significant enriched KEGG pathways.

## DISCUSSION

### Transcriptome analysis of hypoxia adaptation genes in PASMCs under hypoxia and normoxia

Yaks have lived at high altitudes for generations. These animals are able to adapt to the low-oxygen surroundings of the plateau, and pass this adaptability on to the next generation. In our study, PASMCs were cultured and their genetic variations were analysed using RNA-seq technology to grant a theoretical foundation for the adaptability of plateau animals. The characteristics of the PASMCs transcriptome under hypoxic and normoxic conditions were discussed, the differential genes of PASMCs were obtained, and the effects of those differential genes on the adaptive growth of yaks in the plateau environment were studied. PASMCs are necessary to preserving vascular morphology and tension. Under normal physiological conditions, these cells exhibit low proliferation, low migration and strong contractile forces. Under normal circumstances, PASMCs also possess a contractile phenotype; however, when they are stimulated by biochemical substances and machinery, they change to a secretory phenotype, which shows decreased contractile force, enhanced migration and proliferation ability, and an enhanced ability to secrete ECM. Therefore,

in this study, the characteristics of the transcriptome of PASMCs under hypoxic and normoxic conditions were discussed. Bioinformatics shows that gene functions and their relationships can be described from three aspects: biological processes, cell composition, and molecular function. GO functional classification annotation revealed that the DEGs identified were enriched in the top 30 GO categories. The DEGs in biological processes were mainly enrich in: lipid metabolic process, lipid biosynthetic processes, nucleoside metabolic processes, and glycosyl compound metabolic processes. The DEGs in the cellular component (CC) were mainly concentrated in supramolecular complex, supramolecular polymers, supramolecular fibres, and extracellular regions. Finally, the DEGs in molecular functions were mainly enriched in: iron-ion binding, the signalling receptor binding, hormone activity, and molecular functions. The KEGG enrichment pathway results showed that genes with extremely significant down-regulation differences were mainly enriched in the PPAR signalling pathway and fatty acid metabolism. Differential genes in the PPAR signalling pathway (3-hydroxy-3-methylglutaryl-coa synthase 1 (*HMGCS1*), stearoyl-CoA desaturase (*SCD*), etc.) were screened and discussed. Genes with massive up-regulation variations were primarily enriched in staphylococcus aureus infection (C3, C2 and C1s), ECM-receptor interaction and differential genes of ECM-receptor interaction (integrin alpha11 (*ITGA11*), etc.) were selected for discussion.

In this study, a GO enrichment evaluation confirmed that the up-regulated DEGs identified were enriched in the REDOX equilibrium. The REDOX status in VSMCs is determined by reactive oxygen species (ROS), oxidation products and antioxidants. ROS are produced by the mitochondria (*Tuder et al., 2013*; *Gao, Chen & Raj, 2016*; *Sanyour et al., 2020*). In muscle cells, ROS-regulated processes include calcium homeostasis, transcriptional regulation (*Hu et al., 2017*), response to hypoxia, and activation of apoptotic pathways. The downregulated genes were enriched in steroid biosynthesis, such as cholesterol, squalene 2,3-epoxide and cholecystalciferol biosynthesis. Cholesterol coordinates the migration and adhesion of VSMCs to unique ECM proteins, regulates cell hardness, and cytoskeletal orientation, thereby affecting cell biomechanics (*Yang et al., 2017*). The pathways of glucose catabolism in the nucleoside metabolic process mainly include aerobic oxidation, anaerobic glycolysis, the pentose phosphate pathway, and vascular calcification. This finding is in line with results of the existing study, indicating that the effects are reliable.

The KEGG pathway database is one of the most commonly used databases for annotating biological processes at the molecular level. One of the most significantly down-regulated pathways of differential gene expression in the KEGG pathway analysis was the PPAR signalling pathway. The diversity and underestimated conserved levels of PPAR genes may lay the foundation for tumour metabolism, immunity and hypoxic survival in plateau animals. Studies have found that hypoxia further aggravates the disease phenotype of tumour subtypes with abnormal PPAR signalling (*Chang & Lai, 2019*). In this study, seven differentially down-regulated genes (*HMGCS1*, *SCD*, etc.) were identified in the PPAR signalling pathway. Among these, *HMGCS1* is a novel cancer marker. *Zhou et al. (2020)* observed that *HMGCS1* increases the proliferation, migration, and invasion of colon cancer cells, and inhibiting *HMGCS1* significantly reduces the proliferation of colon cancer

cells. In a study on breast cancer, *Walsh et al. (2020)* found that the down-regulation of *HMGCS1* can limit cancer stem cells (CSCs) and identified the pharmacological inhibition of *HMGCS1* as a potential treatment for breast cancer patients. In addition, *HMGCS1* down-regulation in HUVEC reduces cell migration and proliferation (*Ying et al., 2021*). Inhibiting of *HMGCS1* works as a treatment for cancer because down-regulating *HMGCS1* prevents the proliferation and migration of the majority of cancer cells. *SCD* is a key enzyme involved in the biosynthesis and legislation of unsaturated fatty acids. One function of most cancers cells is a trade in lipid composition, in which monounsaturated fatty acids are notably enriched. Increased *SCD1* expression has been found in many cancer cells, and *SCD1* activity has been found to have a direct influence on the tumourigenic pathway (*Parajuli et al., 2017*). *SCD* has become a recent therapeutic target for curing many diseases, and inhibiting *SCD* plays a large role in treating most cancers (*Schnittert et al., 2019*). Compounds that inhibit *SCD1* have been developed and medical trials have been conducted, but inhibition of this gene by hypoxia has not been reported. In this experiment, both *HMGCS1* and *SCD* were significantly down-regulated in PASMCs under hypoxia, indicating that hypoxia could inhibit the expression of *HMGCS1* and *SCD*, but the specific mechanisms behind this result need further verification. The long-term survival ability of plateau animals in a low-oxygen environment is likely directly related to the down-regulation of *HMGCS1*, *SCD*, and other genes, but this conclusion needs further verification. ECM-receptor interaction is one of the pathways that significantly enriches up-regulated genes. ECM is a complicated combination of macromolecules that play vital functions in tissue and in organ morphogenesis and the renovation of cell and tissue shape and function. In addition, integrins serve characteristic as mechanical receptors, offering a force-transfer bodily connection between the ECM and the cytoskeleton. In this study, eight differentially expressed genes were significantly up-regulated (*ITGA11*, *FN1* etc.). *ITGA11* is a collagen receptor and a coding gene of the integrin family. Himalaya Parajuli (*Parajuli et al., 2017*) found that *ITGA11* is normally overexpressed in the stroma of squamous carcinoma of the head and neck and is positively correlated with $\alpha$- clean muscle motion expression (*Stribos et al., 2017*). In addition, *Schnittert et al. (2019)* reported the overexpression of *ITGA11* in the associated fibroblasts in the stroma of pancreatic ductal adenocarcinoma, which became a target for interstitial therapy. Many studies have shown that *ITGA11* is involved in migration, epithelial-mesenchymal transformation, invasion, and metastasis in certain cancers. Thus, *ITGA11* is a promising biomarker and therapeutic target and performs an essential function in cell proliferation, migration, differentiation, and tumour invasion and metastasis. Prior results align with those the of the present study, indicating that the results are reliable. Whether *ITGA11* up-regulation prevents migration of PASMCs and is the key to the adaptive and healthy growth of yaks on the plateau requires further verification. In this study, the differential genes of PASMCs under hypoxic and normoxic conditions were evaluated to provide the basis for studying the adaptive growth of yaks in plateau areas. *FN1* is a high-molecular-weight glycoprotein that exists in the animal ECM and is a core component of complex biological functions and participates in a variety of cell biological processes, including fibrosis (*Liu et al., 2020*) and other diseases. *FN1* is additionally broadly expressed in the ECM of a variety of tumours, such as osteosarcoma,

leiomyosarcoma, and gastric cancers (*Tracz-Gaszewska & Dobrzyn, 2019*). *FN1* is secreted *via* fibroblasts, vascular endothelial cells, liver cells, and VSMCs to alter cell adhesion, proliferation, differentiation, cell morphology maintenance, cell migration promotion, ion exchange, and sign transduction. *FN1* protein overexpression was reported in a recent study, resulting in the deposition of the *FN1* protein in the cytoplasm of cells, ultimately affecting the normal functional activities of cells and leading to fibrosis. The more extreme the bladder fibrosis, the greater the stage of *FN1*. Inhibiting *FN1* synthesis can reduce the fibrosis process of bladder smooth muscle cells (*Skrypek et al., 2021*). *ITGA11* and *FN1* were also two of the DEGs randomly selected for verification and the qPCR results were the same as those in this study, indicating the reliability of the results.

## Proteome analysis of hypoxia adaptive proteins in PASMCs under hypoxia and normoxia

A functional enrichment analysis of DEPs was performed using GO and KEGG, in which the HIF-1 signalling pathway, biosynthesis of secondary metabolites, central carbon metabolism in cancer, and glycolysis/gluconeogenesis extract had six important DEPs (phosphoglycerate kinase (*PGK*), hexokinases (*HK*), lactate dehydrogenase (*LDH*), phosphoglycerate Mutase (*PGAM*), phosphofructose kinase(*pfkA*) and pyruvate dehydrogenase kinase (*PDK1*) (Table 4). PGK is an intracellular protein-and energy-producing glycolytic enzyme that catalyses the reversible transfer of 1, 3-diphosphoglycerate and ADP to produce 3-phosphoglycerate and ATP (*Li, Zhang & Yao, 2019*). Under hypoxic conditions, HIF-1 $\alpha$ up-regulates the expression of *PGK*, provides energy for the glycolysis pathway, and participates in angiogenesis to meet the body's needs and help it adapt to the external environment (*Willson et al., 2022*; *Duncan et al., 2022*). *HK* converts glucose to glucose-6-phosphate and is a key enzyme in regulating glycolysis. The up-regulation of *HK* under hypoxia enhances glycolysis, autophagy and epithelial-mesenchymal transformation (*Chen et al., 2018*). The up-regulation of hexokinase can promote glycolysis by binding to the mitochondrial outer membrane and providing ATP for cell metabolism (*Hong et al., 2021*). *LDH* can be divided into LDHA and LDHB subtypes. *LDHA* regulates TME through HIF signalling during hypoxia. Both HIF-1 $\alpha$ and *HIF-2 $\alpha$* interact with HRE-D in the *LDH* promoter to regulate *LDH* levels. LDH provides information on glycolysis levels and cellular metabolic capacity (*Khan, Nordback & Sand, 2013*; *Hou et al., 2020*). *PGAM* is a key enzyme involved in glycolysis that can promote glycolysis by converting 3-phosphoglycerate into 2-phosphoglycerate. HIF can induce *PGAM* expression and combine with hypoxia reaction elements in the promoter region to exert transcriptional regulation, ultimately leading to an increase in *PGAM* activity (*Mikawa et al., 2020*). *pfkA* is a rate-limiting enzyme for enzymes involved in the cell glycolysis metabolic pathway and catalyses fructose-6-phosphate phosphorylation into fructose 1, 6 diphosphate. Recent research has shown that pfkA expression is up-regulated in a low-oxygen microenvironment to promote cell metabolism and supply electricity to the body. *PDK* is a necessary enzyme that, performs an essential function in regulating glucose and fatty acid metabolism in the body (*Di et al., 2019*). Notably, HIF-1 $\alpha$ can up-regulate *PDK1* and promote glycolysis in tumour cells (*Slominski et al., 2014*; *Ognibene et al., 2017*).

**Table 4  Screening of important DEPs.**

| DEPs name | *P*-value | UP/DOWN | Pathway |
|---|---|---|---|
| PGK | 0.00004383995 | UP | |
| HK | 0.00002034812 | UP | |
| LDH | 0.00001463732 | UP | HIF-1 signaling pathway, Biosynthesis of secondary metabolites, Central carbon metabolism in cancer, Glycolysis/gluconeogenesis |
| PGAM | 0.000007788804 | UP | |
| pfkA | 0.000002953298 | UP | |
| PDK1 | 0.00005644263 | UP | |

## Combined analysis of hypoxia adaptive genes and proteins by PASMCs under hypoxic and normoxic conditions

Analysing mRNA levels or protein levels alone does not provide enough information to fully understand gene expression. Therefore, in this study, the differential genes and proteins associated with yak hypoxia adaptability were explored using a combined analysis. A correlation analysis revealed 109 differentially expressed genes and proteins in PASMCs under hypoxic and normoxic conditions. In addition, there were also genes and proteins with low correlation in the results, because the transition from transcriptome to proteome goes through the translational regulation stage. In a previous study, Schwanhausser pointed out that protein abundance in cells is mainly controlled by the level of translation, and proteins are significantly different from mRNA in terms of half-life, synthesis rate, and quantity (*Schwanhäusser et al., 2011*). Through practical enrichment and evaluation of DEP associated genes using GO and KEGG (Table 5), 71 genes with the same expression trend in both the proteome and transcriptome were identified; these genes were enriched in the hypoxia adaptation-related pathway (HIF-1 signalling pathway, glycolysis and gluconeogenesis, central carbon metabolism in cancer, PPAR signalling pathway, AMPK signalling pathway, and cholesterol metabolism) with nine overlapping genes found in this pathway (*PGAM1*, *PGK1*, triosephosphate isomerase 1 (*TPI1*), *HMOX1*, insulin-like growth factor 1 receptor (*IGF1R*), oxidised low-density lipoprotein receptor 1 (*OLR1*), *SCD*, fatty acid- binding protein 4 (*FABP4*), and low-density lipoprotein receptor (*LDLR*). Two genes were randomly chosen from these nine for RT-qPCR verification, and the results aligned with the sequencing results, indicating the accuracy of these results (Fig. 7). *TPI1* is a key enzyme in glycolytic metabolism, and is regulated by HIF-1 $\alpha$ in hypoxic microenvironments. The expression of *TPI1* was significantly down-regulated by silt HIF-1 $\alpha$. In addition, Zuo et al. found that *TPI1* co-localises with HIF-1 $\alpha$ and an interaction may exist between HIF-1 $\alpha$ and *TPI1* (*Lei et al., 2022*). Haem oxygenase 1 (*HMOX1*) is a downstream target gene of HIF-1 $\alpha$, which is transcriptionally regulated by way of HIF-1 $\alpha$ in hypoxic conditions (*Chillappagari et al., 2014*). HIF-1 $\alpha$ binds to the hypoxic response factor of the *HMOXl* gene and regulates the expression of *HMOX1* in response to hypoxia. The HIF-1 $\alpha$ /HMOX1 pathway has been verified to be necessary in preventing lung damage (*He et al., 2018*; *Han et al., 2020*; *Shi et al., 2021*); and researchers have discovered that activating the HIF-1/HMOX1 signalling pathway can improve LPS-induced acute lung injury by boosting survival rate, reducing inflammation, and reducing oxidative stress. In addition, *Rashid et al. (2020)* concluded that *HMOX1* is a hypoxia response

gene in fish that plays a significant role in the hypoxia tolerance of fish. *IGF1R* plays an important role in tumour cell proliferation and apoptosis (*Al-Saad et al., 2017*). In recent years, relevant research has focused on the dynamic balance between *IGF1R* ubiquitination and deubiquitination (*Al-Saad et al., 2017*) with both performing key regulatory functions determining the survival or mortality of tumours. Research shows that the expression level of *IGF1R* increases hypoxia, which may lead to the synergistic stimulation of cell migration by the paracrine IGF1/ IGFIR signalling pathway (*Yu et al., 2015*). *IGF1R* was also found to be down-regulated *in vivo* as the main component of the *IGF* signalling pathway, which is important to cell life activity (*Yu et al., 2015*). *OLR1* was first found in bovine aortic endothelial cells by *Sawamura et al. (1997)* and is usually involved in regulating the metabolism of fats in the liver and mammary glands. *OLR1* is also related to the with the storage of triacylglycerol, and is generally highly expressed in the lungs, liver, and adipose tissue (*Sun, Liu & Zhang, 2009*; *Catar et al., 2022*). In recent years, *OLR1* research has mainly focused on cardiovascular and metabolic diseases, such as atherosclerosis and diabetes (*Mohammed et al., 2022*). In addition, several studies have shown that OLR1 may also be involved in the incidence and improvement of cancer, in particular tumour metastasis (*Jiang et al., 2019*). However, research on *OLR1*'s role in hypoxia adaptation has not been conducted. *SCD* is a key rate-limiting enzyme for fatty acid synthesis and fat deposition. Hypoxia has been found to saturate fatty acids by inhibiting *SCD*. Although oxygen ($O_2$)-dependent *SCD* enzymes are important for cell survival, their activities can be hypoxia-limited. Therefore, hypoxia causes an accumulation of precursors to saturated fatty acids, which causes the endoplasmic reticulum (ER) membranes to rupture causing cell apoptosis. The provision of exogenous unsaturated lipids can reduce the toxicity caused by saturated fatty acids, indicating that lipid uptake is a key mechanism for preserving a stable intracellular environment in hypoxic cells and lowering cell viability in the absence of exogenous lipid supply. *FABP4* belongs to the family of intracellular lipid-binding proteins. Long-chain fatty acids can bind to *FABP4* proteins, which are involved in fatty acid uptake, transport, and metabolism. During hypoxia, *FABP4* content increases due to the fatty acid energy supplied. The main function of lipid-binding proteins is to actively promote lipid transport to a specific area of the cell, and participate in extracellular autocrine and paracrine effects. Lee discovered that HIF-1 $\alpha$ transcription-activated *FABP4* expression multiplied under hypoxic conditions. The combined activation of HIF1a and HIF1b by HIF1a enhanced *FABP4* promoter activity, and hypoxia-induced *FABP4* expression was significantly decreased under HIF-1 $\alpha$ inhibitors (*Ackerman et al., 2018*). *FABP4* is related to endoplasmic reticulum stress-related apoptosis in more than one context. For example, *FABP4* mediates mesangial cell apoptosis through ER stress in diabetic nephropathy. In liver cells, exogenous *FABP4* causes ER stress and apoptosis. Additionally, *FABP4* knockdown reduces the negative effects of hypoxia/reoxygenation by reducing ER stress-mediated apoptosis (*Lee et al., 2017*). *LDLR* is a membrane mosaic protein involved in the uptake and elimination of endogenous cholesterol. Lipid metabolism is involved in the prevalence and improvement of hypoxic pulmonary hypertension in mice, and the down-regulation of *LDLR* gene expression under hypoxia suggests reduced cholesterol clearance and increased
**Table 5  Enrichment analysis results of genes with similar expression trends in the proteome and transcriptome.**

| Gene set name | Description | Genes in overlap | *P*-value |
|---|---|---|---|
| bta01230 | Biosynthesis of amino acids | 6 | 0.0000000008274853 |
| bta01200 | Carbon metabolism | 5 | 0.0000005137747 |
| bta04144 | Endocytosis | 6 | 0.000001312751 |
| bta00010 | Glycolysis/Gluconeogenesis | 4 | 0.000001899059 |
| bta04145 | Phagosome | 5 | 0.000004067879 |
| bta05205 | Proteoglycans in cancer | 5 | 0.000009315274 |
| bta04066 | HIF-1 signaling pathway | 4 | 0.0000164563 |
| bta04140 | Autophagy - animal | 4 | 0.0000447763 |
| bta04979 | Cholesterol metabolism | 3 | 0.00004547926 |
| bta04913 | Ovarian steroidogenesis | 3 | 0.00009097859 |
| bta05230 | Central carbon metabolism in cancer | 3 | 0.000114256 |
| bta03320 | PPAR signaling pathway | 3 | 0.000213858 |
| bta00100 | Steroid biosynthesis | 2 | 0.000342524 |
| bta04152 | AMPK signaling pathway | 3 | 0.00065382 |
| bta04977 | Vitamin digestion and absorption | 2 | 0.000628869 |
| bta00051 | Fructose and mannose metabolism | 2 | 0.000998926 |
| bta00260 | Glycine, serine and threonine metabolism | 2 | 0.001745798 |
| bta04216 | Ferroptosis | 2 | 0.001745798 |
| bta01523 | Antifolate resistance | 2 | 0.002236525 |
| bta04510 | Focal adhesion | 3 | 0.002669934 |
| bta04213 | Longevity regulating pathway - multiple species | 2 | 0.003285116 |

plasma low-density lipoprotein content in hypoxic pulmonary hypertension, suggesting that abnormal lipid metabolism is part of the formation of hypoxic PH (*Gan et al., 2020*).

With the development of RNA-seq transcriptomics technology, there has been an increase in research on the adaptability of yaks to hypoxia. Qiu et al. compared yak and cattle genomes and found that the Adam17, Arg2, and Mmp3 genes related to extracellular environment and hypoxia stress were enriched in the hypoxia response functional category. This suggests that these genes play a role in high altitude environment adaptations (*Qiu et al., 2012*; *Li et al., 2021*). *Guan et al. (2017)* screened HMOX1, PGK1, and HK in a microRNA transcriptome comparative analysis between yak and bovine (*Guan et al., 2017*). *Lan et al. (2018)* and *Ge et al. (2021)* screened genes such as GAPDH in the glycolytic pathway and metabolism related genes such as SLTM, PARP4, CLCN7 in transcriptome experiments of positive selection genes in yak lungs, the results of which were consistent with the screening results in this experiment, indicating that these genes are unique factors in the adaptation process of yaks to hypoxia. Therefore, the results of this study complement existing data to further explore the mechanisms of adaptation to high altitude hypoxia in high altitude mammals.

## CONCLUSIONS

Transcriptome, proteome, and combined analyses were used to identify possible hypoxia-adaptive genes and proteins in PASMCs under hypoxic and normoxic conditions. This study identified overlapping genes with similar expression trends in the transcriptome and proteome, including *PGAM1*, *PGK1*, *TPI1*, *HMOX1*, *IGF1R*, *OLR1*, *SCD*, *FABP4,* and *LDLR*. Our findings suggest that these differentially expressed genes and protein functional classifications are related to the hypoxia adaptive pathways. Future research should focus on the mutual regulation mechanism of the hypoxia-related factors mentioned above. Our results provide insights into the molecular mechanism of yak PASMCs and the adaptability of plateau animals. In addition, our results serve as a reference for the prevention and cure of high-altitude hypoxic ailments in both humans and animals.

### Funding

This work was supported by the National Natural Science Foundation of China [31860687]; the Natural Science Foundation of Gansu Province [21JR11RA024], the Fundamental Research Funds for the Central Universities [31920200004], and the Program for Changjiang Scholars and Innovative Research Team in the University [IRT-17R88]. The funders had no role in study design, data collection and analysis, decision to publish, or preparation of the manuscript.

### Grant Disclosures

The following grant information was disclosed by the authors:
National Natural Science Foundation of China: 31860687.
Natural Science Foundation of Gansu Province: 21JR11RA024.
The Fundamental Research Funds for the Central Universities: 31920200004.
The Program for Changjiang Scholars and Innovative Research Team in the University: IRT-17R88.

### Competing Interests

The authors declare there are no competing interests.

### Author Contributions

- Lan Zhang conceived and designed the experiments, performed the experiments, analyzed the data, prepared figures and/or tables, and approved the final draft.
- Yiyang Zhang performed the experiments, analyzed the data, authored or reviewed drafts of the article, and approved the final draft.
- Juan Zhou performed the experiments, analyzed the data, authored or reviewed drafts of the article, and approved the final draft.
- Yifan Yao performed the experiments, analyzed the data, prepared figures and/or tables, and approved the final draft.

- Rui Li performed the experiments, analyzed the data, authored or reviewed drafts of the article, and approved the final draft.
- Manlin Zhou performed the experiments, analyzed the data, authored or reviewed drafts of the article, and approved the final draft.
- Shuwu Chen performed the experiments, analyzed the data, prepared figures and/or tables, and approved the final draft.
- Zilin Qiao performed the experiments, analyzed the data, authored or reviewed drafts of the article, and approved the final draft.
- Kun Yang conceived and designed the experiments, performed the experiments, analyzed the data, prepared figures and/or tables, and approved the final draft.

## Data Availability

  The raw measurements are available in the Supplemental Files.

## Supplemental Information

Supplemental information for this article can be found online at http://dx.doi.org/10.7717/peerj.14369#supplemental-information.

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
