# Peer review of "Combined transcriptome and proteome analysis of yak PASMCs under hypoxic and normoxic conditions"

_PeerJ, doi:10.7717/peerj.14369_

## Round 0.1 · original submission · Major Revisions

Please address the concerns of all reviewers and revise the manuscript accordingly.

Reviewer 1 ·

Basic reporting

a. The text is mostly comprehensible. I see that a professional editing service was used but there are still issues with scientific clarity.
E.g. “adaptation genes for hypoxia” (line 26), “adaptive growth of the plateau animals” (line 45), “PASMCs are located in the middle membrane” (line 298), “constant pressure of 250 V” (line 131), and the results section on combined transcriptome and proteomics analysis in general (e.g. line 254-256, 265).
Minor grammatical issues, e.g. unnecessary capitalization (line 39-41), unnecessary hyphenation (line 524), spelling mistakes (line 91, table 3, line 307, Fig 4 legend), unexplained abbreviation (line 57).

b. The study takes a reductionist approach to only the study transcriptome/proteome of pulmonary artery smooth muscle cells in normoxic vs hypoxic environment. For completeness, the introduction should probably at least mention there are other anatomical/physiological adaptations that would not be detectable in this study.

c. Insufficient references in introduction and methods. Similar work was not referenced (e.g. https://doi.org/10.1007/s10709-017-0005-8, https://doi.org/10.1186/s12864-021-08044-9). Surprisingly, no previous research on yaks was cited.

d. The manuscript does have the conventional section headings, but description of methods was found in the results section (line 202-215).

e. Some figures do not appear to be publication ready.
Some individual panels were submitted as figures? Figure size / font size varies wildly. E.g. How large is Figure 3 meant to be? The text in Figure 3D is tiny.
Poor formatting, e.g. spaces missing in labels, subscripts not properly used.
Panel labels should be typically at the top left corner.
Most figure legends are too brief and lack sufficient details to interpret the figures.

f. Processed data were submitted but I was unable to find the raw data files.

g. The lack of a typical ‘V-shaped’ in the volcano plot in Figure 5 would suggest the size of the data markers needs to be reduced for display, otherwise there may be oddities with the data.

h. In-house Perl scripts not provided (line 99).

i. What is the genome website (line 103)?

j. I’m not sure what to make of the statistical power calculation (line 117). I’m not familiar with the software you used. Is a fold change of 1 suppose to be equivalent to an ‘effect’ of 2? A statistical power of 1 would be unusually high unless, for example, the effect size is very large.

k. P values cannot be larger than 1. (Tables)

l. Inconsistent data. How many proteins were identified? (7600 at line 221 vs 6859 at line 33).

Experimental design

a. The lack of a control animal that does not adapt to hypoxic environment severely limits how the transcriptome/proteome data can be interpreted, and does not answer the questions that the authors are trying to address. Instead of investigating the genes/proteins that uniquely allow yak PASMCs to adapt to hypoxic conditions, it is simply reporting hypoxia-related genes/proteins found in yak PASMCs.

b. The current study should be praised for its attempt to complement transcriptome data with proteomic analysis and RT-qPCR. Unfortunately, while inconsistency between the methods were reported there was no serious attempt to explain the low level of correlation.

c. LC-MS/MS mentioned in introduction is not actually used in the study (line 72).

d. Methods relating to how the yak PASMC samples were harvested is missing. Details of culturing conditions are sparse (line 84).

e. Is the work based on a single animal? It is unclear in the methods.

f. It is unclear why a different fold change threshold was used for the transcript vs protein analysis. It is not surprising that fewer proteins were identified.

Validity of the findings

a. As previously stated, there is no comparison against PASMCs from an animal that does not adapt to hypoxia, and therefore it would be incorrect to conclusively claim the reported genes/proteins are involved in the yak’s uncommon ability to adapt to hypoxia.

b. The low level of correlation between the transcriptome and proteome data needs to be properly addressed. Potentially, this could be caused by experimental errors (at the bench or in the analysis), or a true and interesting difference between transcript and protein levels.

c. This type of study is great for identification of potentially important genes/proteins en masse, but there may be too much speculation in the discussion section. Most of the discussion reads like a literature review of various genes/proteins in the context of hypoxia.

Additional comments

The scientific aim seems to match the scope of PeerJ and reasonable technical approaches were employed. However, given fundamental flaws in the experimental design, I would not recommend publication of this manuscript in its current form. The manuscript also requires more editorial work before it is ready. Regardless, previous work in the field needs to be cited and discussed.

Reviewer 2 ·

Basic reporting

In the manuscript by Zhang et al., the authors combined RNA-Seq and proteomic analysis to study the gene expression signature of the Yak cells in hypoxic environment. This study identified 109 differentially expressed genes that were positively correlated with differentially expressed proteins, with 77 genes showing the same expression trend. This manuscript is logically clear. However, the authors need to address the following questions.

1. Abbreviations need to be spelled out when being used for the first time in the manuscript. For example, VSMC in line 57 and TMT in line 30.

2. Literature citations are needed for the descriptions from line 58 to 63.

3. Line 179, “A total of” need to be deleted.

4. Line 202 to 216 should go into the Method section instead.

5. The words in Figure 3C are too small to read.

6. In Figure 4, the authors should also include the data from the RNA-seq so that the RNA fold change from RT-qPCR and RNA-seq can be compared side by side.

7. In Figure 6D and 6E, the Go term description is not even complete as some of them ended with "...".

Experimental design

Overall, this study fits the scope of PeerJ. And the research question was well defined. The following question regarding experimental design needs to be addressed.

In line 200, the authors chose 21 differentially expressed genes and verified their gene expressions using RT-qPCR. Why were these 21 genes selected? What are the selection criteria?

Validity of the findings

All underlying data have been provided.

Additional comments

This study fits the scope of PeerJ. However, the authors need to edit the text and figures accordingly.

Reviewer 3 ·

Basic reporting

The research article titled ‘Combined transcriptome and proteomics analysis of yak
PASMCs under hypoxia and normoxia conditions’ is not appropriately written and needs to be revised/edited. References are missing throughout the article and I ask the authors to do proper referencing. Without references it is not possible to validate the statements given in the article. Methods, Result and Discussion sections are vaguely written and is difficult to follow. Also, the authors have mixed contents of one section with another which needs to be corrected. The content of each section needs to be structured and organized in a proper way. I ask the authors to edit the English language so that readers can easily understand. There are many corrections that should be addressed throughout the paper. Please find below few of my concerns among many that needs to be addressed in the article. I would like to review this article once the authors modify the paper.

Experimental design

See section below

Validity of the findings

See section below

Additional comments

1. Line 57- please expand VSMCs. Abbreviation has to be expanded when it comes first time in the text along with short form in the brackets. Follow exactly how you wrote ECM in line 63.
2. Line 91- What is MLEP tube?
3. Line 126- What is SDT buffer- please expand and write components in brackets
4. Line 126- What is appropriate amount? Please write specific amount/volume. How much cells were used for protein extraction.
5. Line 129- Which method was used for protein quantification?
6. Line 131-Constant pressure of 250 V? Please correct.
7. Line 202 to 216 is included in ‘Results’ which is actually part of Methods section.

---

## Round 0.2 · Major Revisions

Please address concerns of the reviewer #1 and revise the manuscript accordingly. Since it was indicated that the English language must be improved. please make sure that linguistic issues are fixed.

Reviewer 1 ·

Basic reporting

1. The clarity and grammar of the English is much worst after the revision. I had trouble understanding.
2. The authors said there are there few studies on the transcriptome of yak lung tissue and ignored the references I suggested (Line 55-57). Is there a reason they are not included? They look relevant and I list them here:
a. Ge, Q., Guo, Y., Zheng, W. et al. A comparative analysis of differentially expressed mRNAs, miRNAs and circRNAs provides insights into the key genes involved in the high-altitude adaptation of yaks. BMC Genomics 22, 744 (2021). https://doi.org/10.1186/s12864-021-08044-9
b. Qiu, Q., Zhang, G., Ma, T. et al. The yak genome and adaptation to life at high altitude. Nat Genet 44, 946–949 (2012). https://doi.org/10.1038/ng.2343
c. This article is in the reference list but missing citation in-text.
Lan, D., Xiong, X., Ji, W. et al. Transcriptome profile and unique genetic evolution of positively selected genes in yak lungs. Genetica 146, 151–160 (2018). https://doi.org/10.1007/s10709-017-0005-8
3. After the revision, some of the figures are tiny and hard or impossible to read. From what I can read, the legend for Figure 3 is incomplete. Should there be 21 data points in Figure 3F?
4. Authors indicated original data were re-uploaded. I see the processed data, but I still could not find the raw data?
5. Line 88 and 108. Please clarify what methods you followed when citing others.
6. I did not understand the author’s reply on their statistical power calculations. But I have no further comments if the authors are confident in their calculations.

Experimental design

1. It is still unclear whether the work is based on cells from on a single animal.

Validity of the findings

I’m afraid I was not able to understand the essence of the author’s long rebuttal. Let me be more explicit in my comments.
1. Please add discussion (1-2 sentences) and references on low level of correlation between transcriptome and proteome to the article.
2. I stand by that a lack of a control animal (e.g. cattle) does not allow you to conclude these are the genes/proteins that are "unique" for the adaptation of yak to hypoxic environments. Please see Lan et al (2018), Ge et al. (2021) and Qiu (2012) listed above.
But on reading the revision, I realized although the authors suggest this multiple times throughout the manuscript, they do not state this with certainly.
Therefore, I recommend the author add a few sentences to properly address this in the discussion and avoid any misunderstanding. Especially since they already plan to perform similar experiments with ‘maladapted animals’ for comparisons which would address this specific point.
3. The authors should compare their results to the references I listed above.

Reviewer 2 ·

Basic reporting

The authors have sufficiently addressed my concerns.

Experimental design

NA

Validity of the findings

NA

---

## Round 0.3 · accepted · Accept

Authors addressed the remaining issues pointed by the reviewer. Revised manuscript is acceptable now